# Diet Diversity of the Fluviatile Masu Salmon, *Oncorhynchus masou* (Brevoort 1856) Revealed via Gastrointestinal Environmental DNA Metabarcoding and Morphological Identification of Contents

**DOI:** 10.3390/biology13020129

**Published:** 2024-02-18

**Authors:** Lijuan Li, Xuwang Yin, Qianruo Wan, Dilina Rusitanmu, Jie Han

**Affiliations:** 1Ministry of Education Key Laboratory for Biodiversity Science and Ecological Engineering, College of Life Sciences, Beijing Normal University, No. 19 Xin Jie Kou Wai Avenue, Beijing 100875, China; lilijuan17@163.com (L.L.); lucywan1104@126.com (Q.W.); delara0105@sina.com (D.R.); 2Liaoning Provincial Key Laboratory for Hydrobiology, College of Fisheries and Life Science, Dalian Ocean University, No. 52 Hei Shi Jiao Avenue, Dalian 116000, China; yinxuwang@dlou.edu.cn

**Keywords:** conservation, fish, prey, trophic ecology, next-generation sequencing

## Abstract

**Simple Summary:**

Masu salmon (*Oncorhynchus masou*), a commercially important fish endemic to the North Pacific Ocean, received national second-level protected animal status in China in 2021. Employing gastrointestinal tract environmental DNA (GITeDNA) metabarcoding and traditional morphology, this study explored the diet of fluviatile Masu salmon. The results showed a diverse prey composition, with larger fish exhibiting a significant preference for terrestrial invertebrates. The fluviatile Masu salmon acts as a crucial link between aquatic and terrestrial food webs, emphasizing the need for conservation efforts in riparian habitats. This study recommends combining GITeDNA metabarcoding and morphological observation for a comprehensive understanding of fish diet diversity.

**Abstract:**

Masu salmon, *Oncorhynchus masou* (Brevoort 1856), a commercially important fish species endemic to the North Pacific Ocean, attained national second-level protected animal status in China in 2021. Despite this recognition, knowledge about the trophic ecology of this fish remains limited. This study investigated the diet diversity of fluviatile Masu salmon in the Mijiang River, China, utilizing the gastrointestinal tract environmental DNA (GITeDNA) metabarcoding and morphological identification. The results revealed a diverse prey composition, ranging from terrestrial and aquatic invertebrates to small fishes. The fluviatile Masu salmon in general consumed noteworthily more aquatic prey than terrestrial prey. There were much more prey taxa and a higher diet diversity detected by GITeDNA metabarcoding than by morphological identification. GITeDNA metabarcoding showed that larger and older Masu salmon consumed significantly more terrestrial insects than aquatic prey species did, with 7366 verses 5012 sequences in the group of ≥20 cm, 9098 verses 4743 sequences in the group of ≥100 g and 11,540 verses 729 sequences in the group of age 3+. GITeDNA metabarcoding also showed size- and age-related diet diversity, indicating that the dietary niche breadth and trophic diversity of larger and older Masu salmon increased with food resources expanding to more terrestrial prey. Terrestrial invertebrates of riparian habitats play a vital role in the diet of fluviatile Masu salmon, especially larger individuals, highlighting their importance in connecting aquatic and terrestrial food webs. Conservation plans should prioritize the protection and restoration of riparian habitats. This study advocates the combined use of GITeDNA metabarcoding and morphological observation for a comprehensive understanding of fish diet diversity.

## 1. Introduction

The trans-ecosystem linkages and corresponding dynamic processes in food webs are important for adjacent interacting ecosystems. Resource inputs from donor habitats are precious subsidies to the persistence of biological assemblages in recipient habitats [1]. However, intersystem connections rarely receive the same level of attention in studies as structure and dynamics of food webs within a single ecosystem [2]. Freshwater drift-feeding fishes play a critical role in trans-ecosystem linkages between aquatic and terrestrial food webs, because they consume both aquatic and terrestrial prey [3,4,5]. As the body develops and grows, fish dietary shifts are common [6,7]. With increasing energy requirements and decreasing morphological constraints, fish can adjust their foraging behavior to find optimal prey [8]. Drift-feeding salmonid fishes exhibit a size- or age-dependent foraging pattern, with larger individuals having a greater proportion of terrestrial invertebrate prey in their diet compared to smaller conspecifics [9,10,11]. Additionally, the diet composition of a fish can reflect the health status of its feeding grounds. Alterations in the diet can function as valuable indicators of environmental changes, helping to assess the overall health of aquatic ecosystems [12,13]. Masu salmon, *Oncorhynchus masou* (Brevoort 1856), is a commercially important fish species endemic to the North Pacific Ocean, with two distinct ecological forms. One is the fluviatile form, which typically inhabits headwaters and maintains territories, while the other is the sea-run form, which migrates downstream, forming schools. After a brief stay in the brackish zone, the sea-run form enters the sea [14,15]. The fluviatile form could play a part in connecting its aquatic habitat and surrounding terrestrial ecosystem, akin to other drift-feeding salmonid fishes [9,10]. Notably, in 2021, it was designated as a nationally second-level protected animal in China. Exploring the diet composition and diversity of the fluviatile Masu salmon can test this hypothesis. It can provide valuable insights into its ecological role, including understanding its position in the food web, examining how it links the aquatic and terrestrial ecosystems through foraging and contributing to the overall ecosystem dynamics [9,10,11]. Knowledge of the diet composition and diversity can further facilitate evaluation of the health status of its habitat, as well as aid in identifying potential threats to this fish. This information allows for targeted conservation measures to protect the species and its ecosystem [12,16]. Insights into the diet of the fluviatile Masu salmon are also essential for successful aquaculture practices. Understanding its nutritional needs can contribute to the development of optimal feed formulations. This, in turn, leads to healthier and more sustainable aquaculture practices [17,18]. However, information about the trophic ecology of Masu salmon is currently scant.

Conventional investigation of fish diets depends mainly on the morphology-based identification of gastrointestinal tract contents with the assistance of microscopic examination [19,20,21]. However, the accuracy of this approach may be limited in cases, such as empty stomachs [22,23], incomplete or digested prey items [24,25] and a lack of experienced taxonomic expertise [26]. The adoption of molecular technology has been shown to be promising in helping to understand fish diet diversity [27,28,29], especially DNA metabarcoding which facilitates identifying digested and incomplete prey, thus helping to draw a more comprehensive picture of fish diets [30]. Berry et al. compared morphological and DNA metabarcoding analyses of diets of eight marine fish species, and found the latter obtained a higher taxonomic resolution, and revealed dietary items not previously recorded [30]. Sakaguchi et al. successfully identified a higher number of prey taxa of juvenile chum salmon, *Oncorhynchus keta*, by using DNA-based analysis than by morphological observation [31]. However, they inferred that DNA-based analyses probably overestimated the richness of the prey taxa due to the detection possibly including secondary prey [31].

In this study, we investigate diet of the fluviatile Masu salmon through both the gastrointestinal tract environmental DNA (GITeDNA) metabarcoding and morphological identification of contents. Our aims are to determine the diet composition and diversity and thereby understand the ecological role of this fish. Additionally, we aim to test the effects of body size and age on its diet composition and diversity. Lastly, we seek to evaluate the effectiveness of the two study methods in fish dietary analysis.

## 2. Materials and Methods

### 2.1. Sample Collection and Basic Measurements

A total of 31 specimens of fluviatile Masu salmon were collected from the Mijiang River (40°0′2.35″ N~43°12′4.72″ N, 130°9′39.75″ E~130°24′15.42″ E), a tributary of the Tumen River in northeastern Asia (Figure 1). Individuals were captured using a bottom trawl or handheld nets in July 2020. Total length (TL) and total weight (TW) were measured, ranging from 12.2 cm to 26.3 cm and 23.3 g to 248.2 g, respectively. The age of fish was determined to range from 1+ to 3+ by examining scales from its body side above the lateral line and behind the dorsal fin [32,33,34]. Between 15 and 20 scales were observed and photographed for each fish under a compound microscope (SOPTOP EX20, Sunny Optical Technology (group) Co., Ltd., Yuyao, China). The gastrointestinal tract was freshly dissected before being preserved in a 50 mL plastic tube with absolute ethanol for further laboratory analysis.

### 2.2. Morphological Observation of Gastrointestinal Tract Contents

Prey items within the gastrointestinal tract, including partial ones with intact heads, were identified to the lowest possible taxonomic level possible with taxonomic keys [35,36]. The life stage of each identified prey was recorded to determine whether the prey came from terrestrial or aquatic habitats [35]. Adults and nymphs/larvae of all insect orders, as well as non-insect taxa, were categorized based on their habitat use [9]. Partially digested prey that could not be physically isolated for accurate identification was classified as debris, and the relative volume of the debris in comparison to that of the gastrointestinal tract was measured following the methodology outlined in Pirroni et al. [37].

### 2.3. GITeDNA Extraction and Metabarcoding

An amount of 10 mL of ethanol was filtered through a mixed cellulose esters membrane filter with a 1.2 µm pore size from each of the gastrointestinal tract preservation tubes (REF: RAWP04700, Merck Millipore Ltd., Carrigtwohill, Ireland). Then, the filter was cut into the smallest possible pieces using sterile scissors, which had been immersed in a 70% ethanol solution for a minimum of 30 min prior to use. GITeDNA was extracted from the filter pieces using the DNeasy Blood and Tissue Kit (QIAGEN, Hilden, Germany) with slight modifications [38,39]. A filter filtered by 10 mL of absolute ethanol was adopted as a negative control in the filtering and DNA extraction processes. The extracted DNA was quantified by a NanoDrop One (Thermo Fisher Scientific, Madison, WI, USA) and then diluted 10-fold (to reduce the PCR inhibitors) as the working solution.

Three universal primer pairs from Zeale et al. [40], Meusnier et al. [41] and Folmer et al. [42] were tested for PCR amplification of the partial mitochondrial COI gene in a variety of animal groups with all possible combinations to find the best amplification efficiency for our samples. The primers LCO1490 (5′-GGTCAACAAATCATAAAGATATTGG-3′) [42] and ZBJ-ArtR2c (5′-WACTAATCAATTWCCAAATCCTCC-3′) [40] were finally chosen to amplify a ~178 bp of COI gene as the metabarcode of Masu salmon GITeDNA. Two sample-specific oligo DNA tags, eight bases each (Appendix A), were respectively attached to the two PCR primers for sequence demultiplexing in the process of raw sequencing data later on [43,44].

The PCR amplification for the metabarcode was performed in a 20 μL reaction mixture, which included 4 μL of DNA template, 0.6 μL (10 μM) of each primer, 10 μL of 2 × Rapid Taq Master Mix (Nanjing Vazyme Biotech Co., Ltd., Nanjing, China), 3.8 μL of ddH_2_O and 1 μL (0.4 μg/μL) of Bovine Serum Albumin (Sigma-Aldrich, Bayswater, VIC, Australia). To increase specificity and sensitivity in the amplification, a touchdown PCR [45] was run as follows: pre-denaturation at 94 °C for 3 min; 16 cycles of denaturation at 94 °C for 30 s, annealing for 30 s at 63 °C, followed by touching down to 55 °C with a decrease of 0.5 °C in each cycle, and extension at 72 °C for 30 s; then 24 cycles of 94 °C for 30 s, 55 °C for 30 s, and 72 °C for 30 s; and a final extension at 72 °C for 10 min. A negative control for PCR with DNA template replaced by ddH_2_O was included to detect any contamination. PCR amplicons were checked by electrophoresis on a 1% agarose gel (Biowest Regular Agarose G-10, Biowest, Spanish). All PCR amplicons were pooled at equimolar concentrations and purified using the TIANgel Midi Purification Kit (TIANGEN, Beijing, China). A library of the pooled PCR amplicons was constructed with the MGIEasy PCR-Free DNA Library Prep Kit (Shenzhen, China) and sequenced on a MGISEQ-2000 platform (BGI-Shenzhen, Shenzhen, China), using a PE150 sequencing read length and PCR-free library type.

### 2.4. Bioinformatics and Statistical Analyses

Raw sequencing data were processed on the platform QIIME2 [46]. Firstly, raw DNA sequencing reads were imported into the QIIME 2 Artifact using the function *tools import*, followed by six steps: (1) demultiplexing the sequence reads, which used the function *demux-paired* of the Cutadapt plugin to detect barcode sequences and assign sequence reads to the sample based on tags attached to the primers; (2) trimming primers, which used the function *trim-paired* of Cutadapt to find and remove primers and tags from sequences of each sample; (3) denoising and dereplicating the paired-end sequences, and filtering chimeric sequences, which were performed using the function *denoise-paired* of the DADA2 plugin [47]; (4) generating the feature table and corresponding feature sequences, which used the functions *summarize* and *tabulate-seqs* of the Feature-table plugin providing a mapping of feature IDs to sequences and clustering unique amplicon sequence variants (ASVs) [48]; (5) matching the resulting ASVs to the Barcode of Life Database [49], which used the Feature-classifier plugin to find supports for the taxonomic classification of features, and sequences were assigned to the lowest possible taxonomic classifications using the method BLAST+ [50]; (6) ASVs with an abundance less than 10 (summed across all samples) or presence in less than 2 samples were filtered from the feature table using the Feature-table plugin, and then a dietary dataset based on DNA metabarcoding was generated using the *barplot* function of the Taxa plugin. Sequences identified as resulted from species of secondary predation or passive ingestion (like Rotifera, Oomycota, as well as potential parasites), or less than the count of sequences in the negative control PCRs, were excluded to reduce the risk of false positives or tag jumps [44,51,52,53].

The frequency of the identified prey items in the gastrointestinal tract as well as the frequency of ASVs from GITeDNA metabarcoding were calculated [54]. All statistical analyses were run under the R version 4.1.1 [55]. Numbers of prey taxa detected by morphological observation and GITeDNA metabarcoding were compared in the R package VennDiagram. Diet diversity indices including the Shannon–Wiener index, Species richness, Pielou’s evenness and Simpson index [56] were calculated using the R package Vegan [57]. Seven Masu salmon specimens, one with an empty gastrointestinal tract and six with only one prey item, were excluded in diet diversity analyses relating to morphological observation. Pearson’s correlation analysis was used to test the significance of correlation between diet diversity indices obtained through morphological observation and GITeDNA metabarcoding in the R package Vegan [57]. The correlations between diet diversity indices and body size as well as age were tested by generalized linear regression using the function *ggscatter* in the R package Ggpubr. To further investigate the effects of body size and age on diet, fishes were divided into two TL groups (<20 cm and ≥20 cm), two TW groups (<100 g and ≥100 g) and three age groups (1+, 2+ and 3+). The Wilcox test in the R package Ggsignif was run to exam any significance in variations of diet types and diversity indices between the divided groups.

## 3. Results

### 3.1. Identification and Classification of Gastrointestinal Contents

In the gastrointestinal contents of the 31 Masu salmon specimens, 298 prey items were isolated and identified as 42 morphospecies sorted into 3 phyla, 4 classes, 13 orders, 27 families, 21 genera and 8 species (Appendix A). Uncertainties happened at the family, genus and species levels due to incompleteness of the prey. Most of the prey items were insects belonging to five orders, i.e., Diptera, Hymenoptera, Ephemeroptera, Plecoptera and Coleoptera, accounting for 31.8%, 20.1%, 14.1%, 9.4% and 6.0% of the 298 individuals, respectively. Prey fishes, exclusively belonging to the order Cypriniformes, also had a higher proportion of 8.4% of the identified individuals (Appendix A). Aquatic prey significantly outnumbered terrestrial prey (Table 1 and Appendix A). In addition, 35.29% of the aquatic prey items were pollution-tolerant species belonging to the families Culicidae and Chironomidae (Appendix A). The relative volume of debris in the gastrointestinal tract varied from 0% to 60% (Appendix A).

A total of 1,029,450 clean sequencing reads were obtained from GITeDNA metabarcoding. These sequences dereplicated into 775 unique ASVs, of which 147 were assigned to the species level with an abundance of 614,890. They were sorted into 3 phyla, 4 classes, 19 orders, 64 families, 81 genera and 45 species (Appendix A). Uncertainties happened at family, genus and species levels as those in the Barcode of Life Database. The most common prey sequences were from aquatic insects belonging to five orders, i.e., Diptera, Lepidoptera, Ephemeroptera, Trichoptera and Coleoptera, with proportions of 33.3%, 24.8%, 16.1%, 6.3%, and 1.3% in the assigned 614,890 ASVs, respectively. Sequences matched to those belonging to the classes Clitellata and Arachnida as well as the phylum Chordata were also present with higher proportions of 8.4%, 3.5% and 0.4%, respectively. The sequences of aquatic prey were far more abundant than those of terrestrial prey (Table 1, Appendix A). Larger Masu salmon (TL ≥ 20 cm, TW ≥ 100 g) consumed significantly more terrestrial prey, while smaller Masu salmon (TL < 20 cm, TW < 100 g, age 2+ and 1+) ate more aquatic prey (Table 1). Furthermore, 23.07% of the aquatic prey sequences were pollution-tolerant taxa belonging to the families Culicidae, Chironomidae and Tipulidae (Appendix A).

The prey items identified through morphological observation shared three common phyla and four common classes with the assigned prey sequences from GITeDNA metabarcoding. But the metabarcoding recorded sequences representing species in more orders, families and genera, and sorted more concrete species than the observed prey items (Figure 2). At the order level, GITeDNA metabarcoding not only detected all the orders identified through morphological observation, but also recorded six additional orders, namely Haplotaxida, Mecoptera, Rhynchobdellida, Sarcoptiformes, Trombidiformes and Salmoniformes. At the family, genus and species levels, the taxa sorted in morphological observation were mostly also revealed in GITeDNA metabarcoding (Figure 2; Appendix A).

### 3.2. Diet Diversity and the Effects of Body Size and Age

The Shannon–Wiener index, Species richness and Simpson index calculated using data from GITeDNA metabarcoding were 1.96, 35.03 and 0.70, respectively, significantly higher than those calculated using data from morphological observation, which were 1.10, 4.37 and 0.55, respectively. In contrast, Pielou’s evenness from morphological observation, with a value of 0.90, was markedly higher than the value of 0.58 obtained from GITeDNA metabarcoding (Figure 3, Appendix A). Furthermore, an analysis of the diversity indices from the two study approaches did not show an obvious correlation (Appendix A).

For the data obtained from morphological observation (Appendix A), neither correlation analysis nor analysis of variations between the divided groups showed a significant impact of size and age on the diet diversity indices (Appendix A). In the case of data from GITeDNA metabarcoding (Appendix A), TL, TW and age were all significantly positively correlated with both the Shannon–Wiener index and Simpson’s index. However, the correlation between size or age and the Species richness was not significant. Moreover, only TL and Pielou evenness showed a significant positive correlation, while the positive correlation between TW or age and the Pielow evenness did not reach a significant level (Figure 4). The diet diversity indices were generally significantly higher for the group with individuals ≥20 cm compared to the group with individuals <20 cm (Figure 5). However, between the two TW groups, there was no significant difference in any of the diet diversity indices (Figure 5). When comparing age groups, except the Pielou’s evenness, which did not show a distinct difference between any pair of the three groups, other diet diversity indices were significantly higher in the age 2+ group than in the age 1+ group. Between age groups 2+ and 3+, the diet diversity indices did not exhibit a significant difference, except for the Species richness, which was significantly higher in the age 2+ group. Between age groups 1+ and 3+, the Shannon–Wiener index and Simpson’s index were significantly higher in group 3+; however, the differences in the Species richness and the Pielou’s evenness did not reach a significant level (Figure 5).

## 4. Discussion

### 4.1. Diet Composition and Diversity of Masu salmon

In this study, through both GITeDNA metabarcoding and morphological identification, Masu salmon’s diet was found to be ranging from aquatic and terrestrial invertebrates to small fishes. However, since the specimen was only collected in July, these results exclusively represent the summer diet composition. Altogether, the composition of aquatic prey in the gastrointestinal tract of Masu salmon was significantly higher than that of terrestrial prey. This is consistent with the results of previous studies on other drift-feeding salmonid fishes, indicating that aquatic prey is the main source of food for them [9,58,59]. However, the composition of terrestrial prey was also higher, constituting 31.54% by morphological observation and 35.46% by GITeDNA metabarcoding of the diet of Masu salmon. The figures are close or even above the high boundary of the documented scope of 14–33% for terrestrial prey in the diets of other drift-feeding salmonid fishes [9,11,59], demonstrating that terrestrial prey are a very important diet subsidy for Masu salmon. As a big predator, Masu salmon should play a certain role in linking its aquatic habitat and surrounding terrestrial ecosystems.

Debris including the uncountable prey items, plants and sand particles occurred in all but two of the gastrointestinal tracts; one was empty and the other was without debris. Previous studies demonstrated that dietary composition of drift-feeding salmonid fishes was mainly pollution-intolerant aquatic insects [31,60,61,62]. However, in this study, both morphological observation and GITeDNA metabarcoding revealed a certain proportion of pollution-tolerant aquatic prey in the diet of Masu salmon belonging to the families Culicidae, Chironomidae and Tipulidae [63,64,65], which might be a warning signal for habitat pollution.

In terms of diet diversity indices, the Shannon–Wiener index, Species richness, and Simpson index exhibited significant increases in values in the GITeDNA metabarcoding approach, while Pielou’s evenness demonstrated an elevated value in morphological observation. The lack of correlation between diversity indices from the two study approaches suggests that the biodiversity patterns observed through GITeDNA metabarcoding and morphological observation may capture distinct aspects of the ecological community.

### 4.2. Size-Dependent Foraging of Masu salmon

When grouping fish based on body size and age, the metabarcoding data showed that larger and older Masu salmon had significantly more terrestrial than aquatic prey, and on the opposite, the smaller and younger Masu salmon had significantly more aquatic than terrestrial prey, suggesting a size-dependent change in the diet of this fish. Size-dependent change in the diet was also found in other drift-feeding salmonid fishes [9,11,59]. Shifts in diet composition can result from changes in morphological and physiological traits, in that the foraging opportunities of small fish are more limited than those of larger conspecifics [6,7,9,10]. This can also explain the higher fraction of terrestrial prey in the gastrointestinal tract of large individuals of Masu salmon. In addition to reducing morphological and physiological limitations, larger drift-feeding fish have more enhanced swimming ability than smaller conspecifics, which is also very helpful for their foraging on surfaces with lower risk [9,66].

Analysis of GITeDNA metabarcoding revealed significant positive correlations between body size and age of Masu salmon and three diversity indices, excluding Species richness, indicating size- and age-related diet diversity. The TL grouping mirrored this pattern precisely. While the age grouping effectively revealed the size- and age-related diet diversity, it was comparatively weaker than the TL grouping. Surprisingly, the TW grouping was the least effective in discerning the size- and age-related diet diversity, with no significant variation found between the two divided groups. The observed size- and age-related diet diversity implies an expansion in dietary niche breadth and trophic diversity for larger and older Masu salmon, with food resources extending to more terrestrial prey. Specifically, subsidies of terrestrial invertebrates from riparian habitats play a crucial role for larger Masu salmon. These terrigenous subsidies not only reduced competition for common trophic resources but also contributed to population persistence [9,22]. Consequently, the protection and restoration of riparian habitats should be included in conservation plans for Masu salmon.

### 4.3. Comparison of the Study Approaches

It is acknowledged that DNA-based techniques are highly efficient and can detect a broader range of prey in fish diets compared to morphological methods [30,31,67]. In this study, GITeDNA metabarcoding, as anticipated, detected more orders, families, genera as well as species in the diet of Masu salmon than morphological identification did (Appendix A). One reason for this discrepancy could be that GITeDNA metabarcoding can identify many digested prey items or debris in the gastrointestinal tract that are unrecognizable by morphological methods, providing more taxonomic information on prey [13,68]. Another factor contributing to the increased prey identification by GITeDNA metabarcoding may be its ability to detect traces of digested prey left in the gastrointestinal tract, as long as proper PCR primers for metabarcode were utilized. However, it should be noted that due to primer non-specificity, there were several prey species observed through morphological observation in this study that did not appear in the data from GITeDNA metabarcoding. Therefore, morphological methods remain valuable for achieving a comprehensive and objective understanding of the diet in fish.

Similar to other methodologies, DNA-based methods have their own limitations. Firstly, distinguishing whether the sequence originates from adult insects or their larvae can be challenging. Especially when the larvae of certain insects are aquatic and the adults are terrestrial, and in cases where both life stages are present in the fish diet, difficulties may arise in differentiating terrestrial and aquatic prey [69,70]. Secondly, there is a certain possibility of a false-positive detection of secondary prey [31,71,72]. While very few secondary prey sequences were detected in the diet of Masu salmon through GITeDNA metabarcoding, efforts to exclude these sequences, such as those from Rotifer and some potential parasites, may not entirely eliminate the chance of recording a small number of secondary prey sequences as prey of Masu salmon. However, it should be noted that secondary prey can, to some extent, provide insights into the energy sources of fish and energy linkages in food webs [71]. The third limitation of the DNA-based analyses is the potential neglect of abiotic components mixed in the diets [73]. For example, microplastics presented in the gastrointestinal tract of a fish can only be identified by morphological observation. Moreover, DNA metabarcoding currently faces the challenge on sequence quantification [70]. Therefore, morphological observation remains valuable for detecting tangible abiotic substances in the gastrointestinal tract of fish and evaluating corresponding environmental issues. We advocate the combined use of both methods, complementing each other, to achieve an objective and comprehensive understanding of fish diet diversity.

Masu salmon specimens in this study were dissected for other research needs; otherwise, we recommend non-invasive sampling, for example, to obtain the gastrointestinal contents of fish by flushing with water using a Seaburg’s pump [74] and then the water can be used as a source of GITeDNA.

## 5. Conclusions

Masu salmon displays a diverse diet, ranging from terrestrial and aquatic invertebrates to small fishes. This dietary variety is size and age related, with aquatic prey identified as the primary food source. Notably, terrestrial invertebrates play a crucial role, particularly for larger fish, acting as significant diet subsidies. Masu salmon, by connecting aquatic and terrestrial food webs, assumes a pivotal role in adjacent ecosystem dynamics. Conservation strategies for this species should prioritize the protection and restoration of riparian habitats. Additionally, GITeDNA metabarcoding proves highly efficient in detecting diet composition and diversity, while morphological observation remains indispensable for identifying tangible prey items and abiotic substances. The combined utilization of both methods enables a comprehensive understanding of the diet diversity of fish.

## Figures and Tables

**Figure 1 biology-13-00129-f001:**
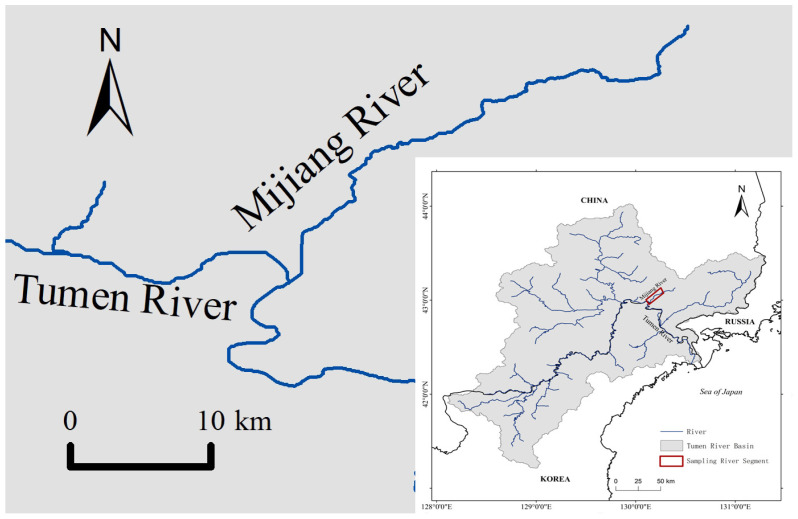
Map showing sampling locality.

**Figure 2 biology-13-00129-f002:**
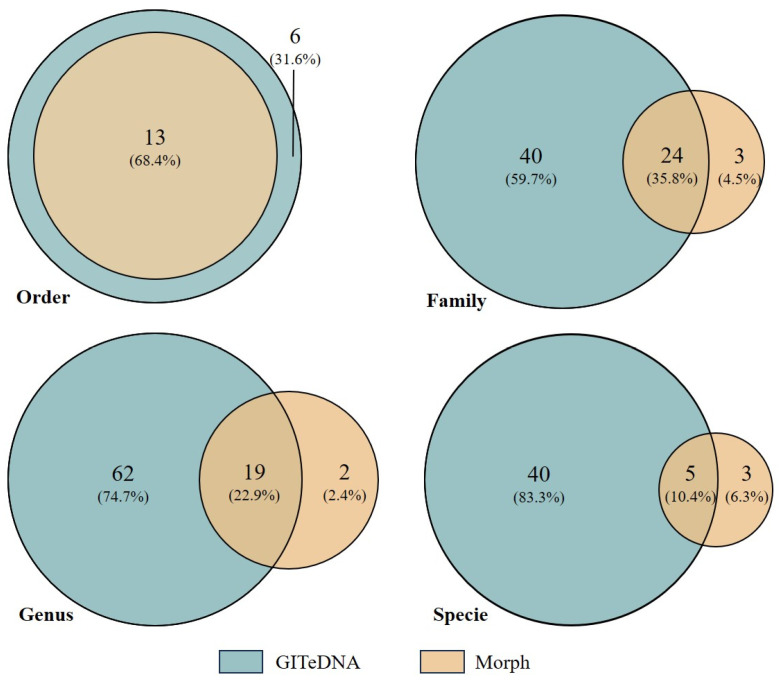
Venn diagrams comparing the numbers of prey taxa detected through the gastrointestinal tract environmental DNA (GITeDNA) metabarcoding and morphologic observation (Morph).

**Figure 3 biology-13-00129-f003:**
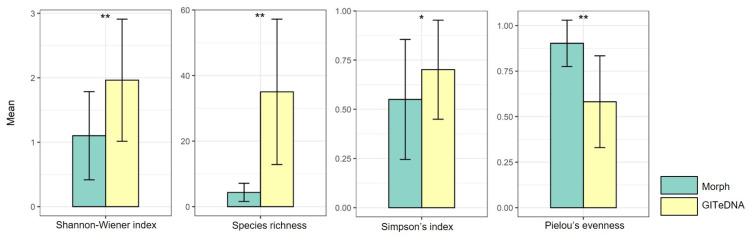
Comparison of diet diversity indices between those calculated using data from morphological observation (Morph) and from GITeDNA metabarcoding. (* *p* < 0.05; ** *p* < 0.01).

**Figure 4 biology-13-00129-f004:**
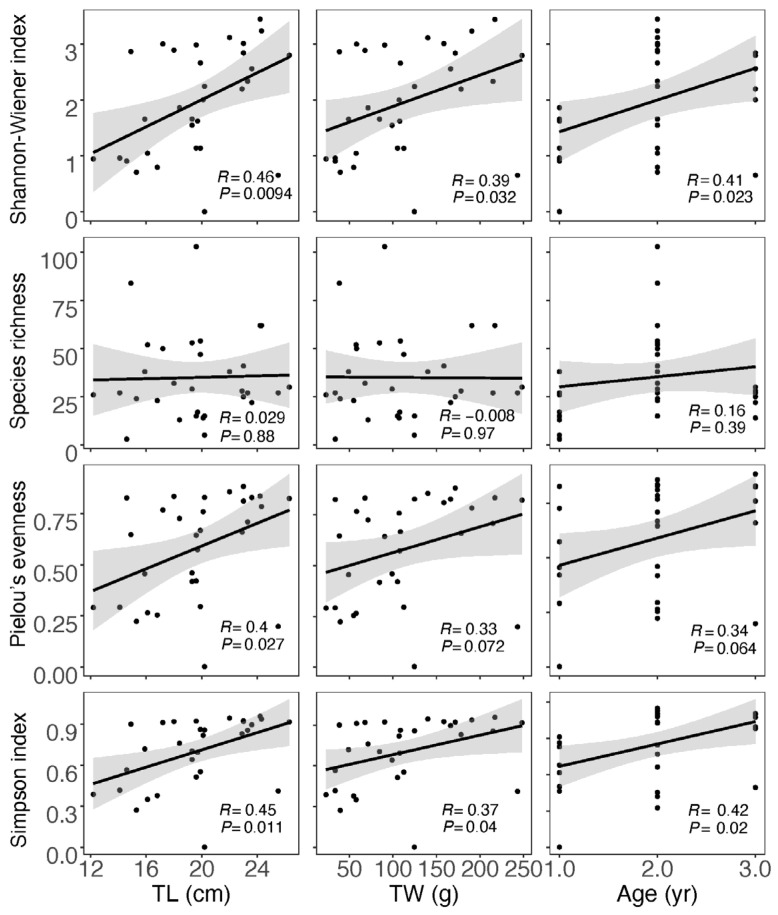
Correlation between diet diversity indices calculated using GITeDNA metabarcoding data and body size and age of Masu salmon. (TL: total length; TW: total weight; *n* = 31).

**Figure 5 biology-13-00129-f005:**
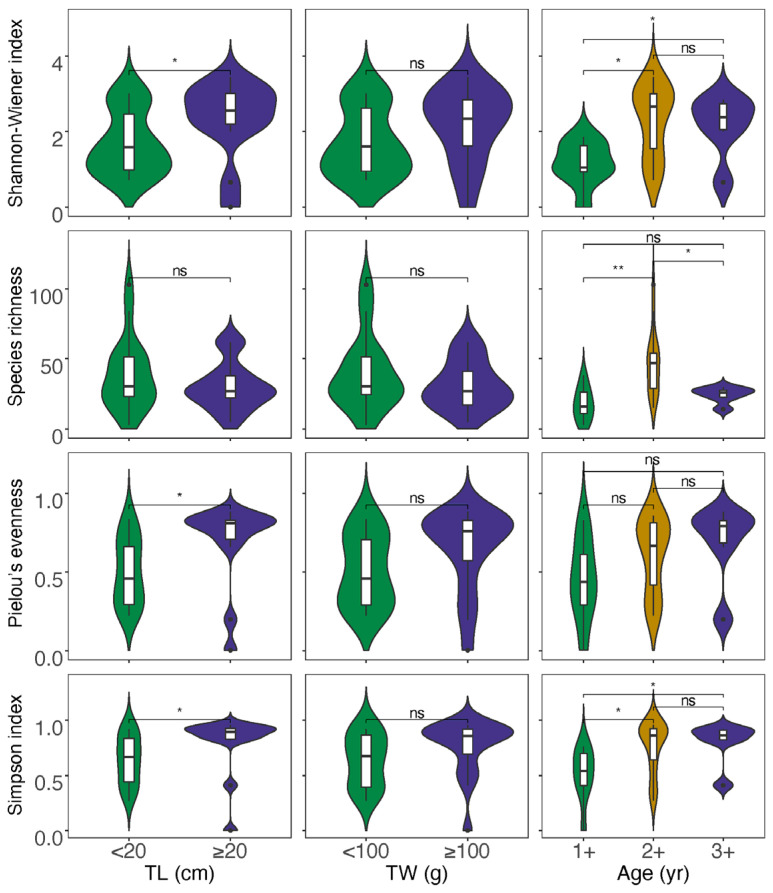
Analyses of variations for diet diversity indices between different size and age groups (TL: total length; TW: total weight; * *p* < 0.05, ** *p* < 0.01; ^ns^
*p* ≥ 0.05).

**Table 1 biology-13-00129-t001:** Composition of terrestrial and aquatic prey in divided Masu salmon groups.

Data		Group	<20 cm	≥20 cm	<100 g	≥100 g	Age 1+	Age 2+	Age 3+	Total
	*n*									
	Prey Type		18	13	14	17	8	17	6	31
GITeDNA (Mean ± sd)	Terrestrial		6793 ± 15,081	7366 ± 25,476	4526 ± 9433	9098 ± 25,422	6080 ± 12,259	4480 ± 13,715	11,540 ± 37,534	7033 ± 19,713
	Aquatic		18,427 ± 19,372	5012 ± 13,530	22,588 ± 20,115	4743 ± 11,806	19167 ± 20,719	14,067 ± 18,877	729 ± 673	12,802 ± 18,198
	*p*		*p* < 0.01	*p* < 0.05	*p* < 0.01	*p* < 0.05	*p* < 0.05	*p* < 0.01	*p* ≥ 0.05	*p* < 0.01
Morph (Mean ± sd)	Terrestrial		4.72 ± 5.67	0.692 ± 1.18	5.5 ± 6.19	1 ± 1.41	5.25 ± 7.15	2.71 ± 3.95	1 ± 1.55	3.03 ± 4.78
	Aquatic		7.22 ± 7.94	5.69 ± 6.40	7.93 ± 8.77	5.47 ± 5.77	4.25 ± 3.65	8.24 ± 8.95	5 ± 4.43	6.58 ± 7.26
	*p*		*p* ≥ 0.05	*p* < 0.01	*p* ≥ 0.05	*p* < 0.01	*p* ≥ 0.05	*p* < 0.01	*p* < 0.05	*p* < 0.01

## Data Availability

All data generated or analyzed during this study are included in this published article (and its Appendix A).

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
