# Peer review of "Diet Diversity of the Fluviatile Masu Salmon, Oncorhynchus masou (Brevoort 1856) Revealed via Gastrointestinal Environmental DNA Metabarcoding and Morphological Identification of Contents"

_biology, 2024, doi:10.3390/biology13020129_

Round 1
Reviewer 1 Report
Comments and Suggestions for Authors
Title: Diet diversity of the fluviatile Masu salmon revealed by the gastrointestinal tract environmental DNA metabarcoding and contents morphological identification
Manuscript ID: biology-2824926
Comments
The authors aimed to investigate the diet composition, diversity and ecological role of this fish, the effects of body size and age of the fish on its diet composition and diversity of fluviatile Masu salmon using both gastrointestinal tract content and environmental DNA metabarcoding techniques. Firstly, the scientific idea of this work is excellent as it discusses two techniques for measuring the gut contents and relates to the body size, age and environment. However, the experiment is not defined properly I also found some descriptions are missing (see details comments and suggestions in the manuscript file). The manuscript could be reconsidered after carefully addressing the following issues and also looking into the manuscript!!

Comments on the Quality of English LanguageMinor English would be required.
Author Response
Dear Reviewer,
We sincerely appreciate your valuable time and effort dedicated to reviewing our manuscript. Your insightful comments and suggestions have been instrumental in refining our work. In response to your feedback, we have made comprehensive revisions to address each of your points. Enclosed is our point-to-point reply, outlining the specific actions taken.
Thank you once again for your constructive feedback, which has significantly contributed to the improvement of our manuscript. We eagerly await your further guidance to ensure the continued enhancement of its quality.
Best regards,
Jie Han, Lijuan Li, Xuwang Yin, Qianruo Wan, Dilina Rusitanmu
College of Life Sciences, Beijng Normal University, Beijing, China
Point-to-point Reply
Comments
The authors aimed to investigate the diet composition, diversity and ecological role of this fish, the effects of body size and age of the fish on its diet composition and diversity of fluviatile Masu salmon using both gastrointestinal tract content and environmental DNA metabarcoding techniques. Firstly, the scientific idea of this work is excellent as it discusses two techniques for measuring the gut contents and relates to the body size, age and environment. However, the experiment is not defined properly I also found some descriptions are missing (see details comments and suggestions in the manuscript file). The manuscript could be reconsidered after carefully addressing the following issues and also looking into the manuscript!!
Response: Thank you for your thoughtful comments. We appreciate your positive acknowledgment of the scientific idea behind our work, focusing on the diet composition, diversity, and ecological role of fluviatile Masu salmon. We also value your constructive criticism regarding defining experiment and missing descriptions. We have carefully reviewed your detailed comments and suggestions in the manuscript file. We understand the importance of defining the experiment more clearly and filling in the missing descriptions. We are committed to addressing these issues comprehensively to enhance the overall quality of our paper.
Your insights are invaluable, and we are grateful for the opportunity to refine our work based on your comments and suggestions. We have thoroughly reconsidered the manuscript and made the necessary revisions to ensure a more robust and well-defined presentation of our research.
If you have any specific point you would like us to focus on or if there are particular areas that require further clarification, please feel free to provide additional guidance. We are eager to improve our paper based on your expertise and suggestions.
Revision of the title
Response: Thank you very much, and the title has been modified in accordance with your constructive suggestion.
2.1. Sample collection and basic measurements
- Total how many fish were collected?
Response: Thanks. A total of 31 fish specimens were collected and this information is presented in the first sentence of Section 3.1 in the results.
- Did the author collected all the specimens at a time or more than once?
The information are not clear.
Response: Thanks. As the fish became rare, the specimens were collected many times in July 2020.
Sometimes, it would not present the whole picture if the samples collected from one station.
Response: The specimens were collected in the Mijiang River (40°0′2.35″N~43°12′4.72″N, 130°9′39.75″E~130°24′15.42″E), and this information is presented in the first sentence of Section 2.1.
4.1 Diet composition and diversity of Masu salmon
If you want to discuss according to the objectives, you should give a separate subheading for your objective 2.
Response: Thank you. We have divided the context of 4.1 into two parts, with the new 4.2 focuses our objective 2.
4.2 Indication of environmental problem
For this section you didn't present any analyses in the results section.
Response: We appreciate your attention to detail and your constructive comments regarding Section 4.2 - "Indication of environmental problem." After careful consideration, we have decided to remove this section from our paper. The environmental problem was mainly inferred from the composition of prey species and we have incorporated this information to revised 4.1.
We understand the importance of aligning the content of the results section with the topics discussed and in light of your comment, we have revised the manuscript to ensure that the discussion section accurately reflects the content of results section, and we have eliminated discrepancies in the presentation.
Reviewer 2 Report
Comments and Suggestions for Authors
English language:
I found some part of MS needs English editing. I will encourage the author to edit the language of this paper.
Plagiarism:
20%. Acceptable but try to reduce below 15%.
Title:
-Please write the scientific name of the species along with the author and year of discovery.
-Please reduce the title texts but carry the same information (Need to rethink)
Summary:
L19: I think this is a huge statement. Rather you focus on what your research can do with the mitigation of these pollution problems or simply need not write it.
Abstract:
An ideal abstract comprised of,
-Introduction
-Objective+/hypothesis/research question
-Methodology
-Results (must contain some values/digits)
-Conclusions
-Implication of your study.
Please follow the structure for shaping the abstract.
L32: Where is the data?
-Research question/problem statement is missing in the abstract section.
-The study area and the study site are not mentioned in the abstract.
Keywords:
-All of your keywords showcased in the keywords section are already in the title. Please replace them with new keywords. Your keywords must be different from your title keywords.
Introduction:
L46: -No hypothesis is discussed
-No research question is raised
-A clear research question/hypothesis is absent in the study.
-Clear research objectives are missing. Please write your objectives according to the journal’s format.
-Implication of the study is absent.
L68: The current problem statement is not strong. Please follow the comment:
Please provide the other studies' niches as well. In which dimension these species were studied? Please mention in this part of your introduction section with some citations, this will help build the problem statement.
L76-78: The citation must appear at the end of the sentence. This is not the sentence style in the introduction section.
L79-82: The citation must appear at the end of the sentence. This is not the sentence style in the introduction section.
L87-89: Please write in a sentence. You do not need to separate the objectives with numbers.
Materials and methods:
L105: Please mention some taxonomic books including insects and local fishes.
L115-116: Mention the paper number/model
L123-124: Insert one space after every word.
L151-169: Please write the citations and no need to add these texts. If you modify some process, you need to write the modification thing.
L170: Use a citation here.
L172-173: Cite their main paper where they declared these indices.
The P value in the table: This P must be in capital and italic.
Results:
-Some sections of the result section need to rewrite.
L200-201: Better to include them in the discussion section.
L203-205: Better to include them in the discussion section.
L206-207: It seems methodology. Please write only results in this section.
L233-234: Is there any other way to represent this Venn diagram? For example, pie, chart, etc. This concept is not matching at all.
L236: Describe the table. We want values here. Your interpretation must be seen in the discussion section.
L239-241: Describe the table. We want values here. Your interpretation must be seen in the discussion section.
L244-246: Describe the table. We want values here. Your interpretation must be seen in the discussion section.
L247-248: Describe the table. We want values here. Your interpretation must be seen in the discussion section.
Discussion:
-The discussion section needs to be rewritten according to your result section.
Sub-headings of results and discussion section must be the same. Here you provided different headings.
Tips for writing discussion:
Firstly, you need to write your result and then interpret it according to the existing facts, and then you can add your arguments to support your study. Follow this structure to write the whole section.
Conclusions:
Write in one paragraph. Your conclusion must be aligned with the objective and hypothesis you build.

Comments on the Quality of English LanguageSome moderate language will be required.
Author Response
Dear Reviewer,
We sincerely appreciate your valuable time and effort dedicated to reviewing our manuscript. Your insightful comments and suggestions have been instrumental in refining our work. In response to your feedback, we have made comprehensive revisions to address each of your points. Enclosed is our point-to-point reply, outlining the specific actions taken.
Thank you once again for your constructive feedback, which has significantly contributed to the improvement of our manuscript. We eagerly await your further guidance to ensure the continued enhancement of its quality.
Best regards,
Jie Han, Lijuan Li, Xuwang Yin, Qianruo Wan, Dilina Rusitanmu
College of Life Sciences, Beijng Normal University, Beijing, China
Point-to-point Reply
English language:
I found some part of MS needs English editing. I will encourage the author to edit the language of this paper.
Response: Thank you for your constructive feedback. We acknowledge your observation regarding the need for English editing in certain parts of the paper. We have carefully addressed and edited those sections to enhance the overall language quality of the manuscript.
Plagiarism:
20%. Acceptable but try to reduce below 15%.
Response: Thank you and we have made every effort to reduce plagiarism detection values.
Title:
-Please write the scientific name of the species along with the author and year of discovery.
-Please reduce the title texts but carry the same information (Need to rethink)
Response: Thank you very much, and the title has been modified in accordance with your constructive suggestion.
Summary:
L19: I think this is a huge statement. Rather you focus on what your research can do with the mitigation of these pollution problems or simply need not write it.
Response: Thank you very much for this constructive suggestion and we agree to delete this sentence after careful consideration.
Abstract:
An ideal abstract comprised of,
-Introduction
-Objective+/hypothesis/research question
-Methodology
-Results (must contain some values/digits)
-Conclusions
-Implication of your study.
Please follow the structure for shaping the abstract.
Response: Thank you very much for your clear guidance regarding the ideal structure of abstract, and we have followed the suggested format for shaping our abstract. Our commitment is to provide a clear and well-organized abstract that encompasses the key elements you have outlined. By adhering to this structure, we aim to enhance the overall clarity and completeness of our abstract. We value your input and are dedicated to incorporating your recommendations to improve the quality of our manuscript.
L32: Where is the data?
-Research question/problem statement is missing in the abstract section.
-The study area and the study site are not mentioned in the abstract.
Response: Thank you for your valuable comment. In the revised version, we explicitly state the research question or problem to provide a more focused and informative abstract. We have also incorporated information about the study area and site in the abstract to enhance the context and understanding of the research.
Keywords:
-All of your keywords showcased in the keywords section are already in the title. Please replace them with new keywords. Your keywords must be different from your title keywords.
Response: Thank you for your guidance. We have replaced the keywords with new terms that differ from those already present in the title. This adjustment aims to enhance the diversity and specificity of our keyword selection, and contributes to a more comprehensive understanding of the research focus.
Introduction:
L46: -No hypothesis is discussed
-No research question is raised
-A clear research question/hypothesis is absent in the study.
-Clear research objectives are missing. Please write your objectives according to the journal’s format.
-Implication of the study is absent.
Response: Thanks for your detailed feedback on the introduction section of our manuscript. We appreciate your insightful comments. In the revised version, we have tried to explicitly state the hypothesis and pose a well-defined research question to guide our study. we have rewritten and present our objectives in accordance with the specified journal guidelines. we have also explicitly discussed the broader implications of our research to provide a more comprehensive context.
L68: The current problem statement is not strong. Please follow the comment:
Please provide the other studies' niches as well. In which dimension these species were studied? Please mention in this part of your introduction section with some citations, this will help build the problem statement.
Response: Thanks a lot for your valuable comment. We appreciate your suggestion to strengthen the problem statement. In response to your comment, we have tried to enhance the introduction by providing a comprehensive overview of other studies' niches and specifying the dimensions in which these species were studied. We have also incorporated relevant citations to support and build upon the problem statement, ensuring a more robust and well-grounded foundation for our research.
L76-78: The citation must appear at the end of the sentence. This is not the sentence style in the introduction section.
Response: Okay, we have moved the citation to the end of the sentence.
L79-82: The citation must appear at the end of the sentence. This is not the sentence style in the introduction section.
Response: Yes, the citation has been moved to the end of the sentence.
L87-89: Please write in a sentence. You do not need to separate the objectives with numbers.
Response: Certainly. We have revised the objectives in single sentences without using numbers.
Materials and methods:
L105: Please mention some taxonomic books including insects and local fishes.
Response: Absolutely. We have added books to the citations that provide comprehensive taxonomic information for invertebrates and local fishes.
L115-116: Mention the paper number/model
Response: Okay.
L123-124: Insert one space after every word.
Response: Done.
L151-169: Please write the citations and no need to add these texts. If you modify some process, you need to write the modification thing.
Response: Thank you for your guidance. It should be our misleading expression and we apologize for confusion. Actually, our raw sequencing data were processed on the platform QIIME2, an open-source microbiome data science platform. Our data processing steps are designed for this study. In the revised version, we have rephrased the sentences to describe more clearly about the data analysis.
L170: Use a citation here.
Response: Okay.
L172-173: Cite their main paper where they declared these indices.
Response: Done.
The P value in the table: This P must be in capital and italic.
Response: Certainly.
Results:
-Some sections of the result section need to rewrite.
L200-201: Better to include them in the discussion section.
Response: Thank you for your guidance. We thus move this sentence to the discussion section for a clarity of context of our manuscript.
L203-205: Better to include them in the discussion section.
Response: Thanks again. As other reviewers suggest excluding the content about microplastics in this article and make the topic more focused, after careful thinking, we agreed. The revised version should more explicit and readable.
L206-207: It seems methodology. Please write only results in this section.
Response: Thank you for this comment. And we have rephrased the sentences to ensure that only results are presented in this section, with a clear separation from the methodology.
L233-234: Is there any other way to represent this Venn diagram? For example, pie, chart, etc. This concept is not matching at all.
Response: Thank you for your thoughtful consideration of the representation of our results. We appreciate your suggestion to explore alternative ways to present the Venn diagram. While we understand the importance of considering various visualization methods, we believe that the Venn diagram is a widely accepted and appropriate choice for illustrating the overlapping sets in our study.
The Venn diagrams effectively compare the numbers of prey taxa detected through GITeDNA metabarcoding and morphologic observation, and use of Venn diagrams aligns well with established conventions in our field. However, we acknowledge your concern and want to assure you that we have carefully considered alternative visualizations. After a thorough evaluation, we find that the Venn diagram best suits the nature of our data and provides a clear and accurate representation of the results.
We value your input and are open to further discussion on this matter. If you have specific reasons or concerns related to the Venn diagram that we may not have addressed, please feel free to provide additional feedback. We are committed to ensuring that our chosen visualization method enhances the overall clarity and understanding of the results.
L236: Describe the table. We want values here. Your interpretation must be seen in the discussion section.
Response: Thank you for your comment and guidance. We have provided a description of the table and presented the relevant values. The interpretation of these values has been discussed in the discussion section.
L239-241: Describe the table. We want values here. Your interpretation must be seen in the discussion section.
Response: Thank you for your comment and guidance. We have added a table S4 and provided a description of the table and presented the relevant values. The interpretation of these values has been discussed in the discussion section.
L244-246: Describe the table. We want values here. Your interpretation must be seen in the discussion section.
Response: Thank you for your comment and guidance. We have added a table S4 which presents the relevant values. The interpretation of these values has been discussed in the discussion section.
L247-248: Describe the table. We want values here. Your interpretation must be seen in the discussion section.
Response: Thank you for your comment and guidance. We have added a table S4 which presents the relevant values. The interpretation of these values has been discussed in the discussion section.
Discussion:
-The discussion section needs to be rewritten according to your result section.
Sub-headings of results and discussion section must be the same. Here you provided different headings.
Tips for writing discussion:
Firstly, you need to write your result and then interpret it according to the existing facts, and then you can add your arguments to support your study. Follow this structure to write the whole section.
Response: Thank you for your helpful tips for writing discussion. We have aligned the sub-headings in the discussion section with those in the results section. Additionally, we followed the suggested structure by presenting the results first and then interpreting them based on existing facts. This approach has been followed by incorporating arguments to support the study throughout the discussion section.
Conclusions:
Write in one paragraph. Your conclusion must be aligned with the objective and hypothesis you build.
Response: Certainly, I will align the conclusion with the objectives and hypotheses established in the study, ensuring coherence and a seamless connection between these elements.
Reviewer 3 Report
Comments and Suggestions for Authors
General comments:
The choice of subject is relevant and important to understanding diet composition and diversity of the fluviatile Masu salmon (Oncorhynchus masou). The study is not just increase the knowledge on the Masu salmon ecological role, but also help assess the healthy status of its habitat, as well as benefit conservation and aquaculture of this fish. To get a comprehensive knowledge about the diet diversity of this fish, the authors used two different kinds of method:
1) the gastrointestinal tract environmental DNA (GITeDNA) metabarcoding,
2) and traditional morphological identification.
With the GITeDNA metabarcoding method, many more prey taxa and higher diet diversity were detected, and larger and elder specimens consumed significantly more terrestrial insects than aquatic prey species. Size- and age-related diet diversity were detected, indicating the dietary niche breadth and trophic diversity of larger and elder Masu salmon.
The manuscript is well writtened and straightforward. In the following, I have attached some comments related to the article draft to contribute for its improvement. Due to its scientific impact, I suggest a minor revision of the manuscript, and after the corrections, I am supporting the publication of it.
Simple Summary and Abstract: The summary of the study's aims is well-designed, and the reader can quickly get to the subject of the investigation. At the end of the abstract and summary, the authors emphasize the importance of Masu salmon as the link between the aquatic and terrestrial food webs, and they mention the protection of Masu salmon habitats. However, there needs to be more information about the conservation status of the species. According to the IUCN database (https://www.iucnredlist.org/species/15319/125852789) and Fishbase.org (https://fishbase.mnhn.fr/summary/242), Masu salmon is an endangered salmonid species in the wild from the Northwest Pacific, therefore emphasizing its conservational state (Endangered) should be necessary – highlighting the importance of this study as well.
Introduction:
L61: „generally inhabits headwaters and often maintains a territory” - this sentence matches 100% with the description from the FishBase. Please rephrase the sentence to avoid plagiarism.
Material and Method:
2.1. Sample collection and basic measurements
In this section, we get informations about the sampling area (Mijiang River), sampling equipment (bottom trawl or 94 handheld nets) and the date of sample collection (July 2020), the estimation of age (scale method). For further diet analysis, the authors dissected the gastrointestinal tract of fish and preserved it in 50 ml 96% alcohol (ethanol).
Fig.1. The map related to the sampling locality needs some modification. Only a tiny part of the map (with sampling segment) is informative for readers. I suggest using this map as a secondary map, which shows the locality of the sampling area, and using a second - main map – that focuses on the sampling segment (the Mijiang River).
See the following article:
https://www.sciencedirect.com/science/article/pii/S1470160X19300019#f0005
2.2. Morphological observation of gastrointestinal tract contents
L108-109: How did you measure the relative volume of the derbies? Can you describe it in more detail?
L109: Did you measure the relative volume only for the debris or other prey items? Please, correct or complement this paragraph according to the question.
L110-112: „Tangible abiotic matters, especially microplastics if present in the gastrointestinal tract were also recorded”. Authors stored the eviscerated GIT in plastic tubes until labratory investigations (see at L99-100), which can result in the contamination of the samples with microplastic. Besides, studying microplastics requires strict terms of preparation. Therefore, I do not recommend reporting microplastic detection in this investigation.
2.3 GITeDNA extraction and metabarcoding
L116-117: „Then, the filter was cut into as small pieces as possible using 116 sterile scissors.” How did you sterilize the scissors? Like used ethanol? Please detail it.
2.4 Bioinformatics and statistical analyses
Please, do not forget to cite the R packages (etc. vegan, Ggpubr).
Like Vegan package: Oksanen J, Blanchet FG, Michael F, Roeland K, Legendre P, McGlinn D, Minchin PR, O’Hara RB, Simpson GL, Solymos P, Stevens MHH, Szoecs E, Wagner H (2020) vegan: Community Ecology Package. R package version 2.5–7.
Results:
3.1. Identification and classification of gastrointestinal contents
In the first paragraphs, the authors list each prey items in percent (%), however, it is not clear that is this the prey-specific volume of each prey items or relative volume? If it is prey-specific volume, please describe the calculation of it in the Material and methods chapter too and recommend you to use and cite the following article, which used the same method:
https://neobiota.pensoft.net/articles.php?id=95680&journal_name=neobiota
L203-205: I reccommend to delete this sentence based on my previous comment.
3.2. Diet diversity and the effects of body size and age
L251: „Between age groups 2+ and 3+, there diet diversity indices” à correct spelling to their
Figure 3. In the figure, there are three asterics on the barplots’ top of the S-W index, Species richness, and Pielou’s evenness, but in the figure caption, I only find 1 and 2 asterics. Maybe this could be a typo error. Can you correct it?
Discussion:
4.2 Indication of environmental problem
L305: According to my previous comments, please consider the subject of microplastic in this study. Without that part, the study is still informative and contains essential scientific information.
Conclusion:
L360-362: See my previous comments related to microplastic.
Author Response
Dear Reviewer,
We sincerely appreciate your valuable time and effort dedicated to reviewing our manuscript. Your insightful comments and suggestions have been instrumental in refining our work. In response to your feedback, we have made comprehensive revisions to address each of your points. Enclosed is our point-to-point reply, outlining the specific actions taken.
Thank you once again for your constructive feedback, which has significantly contributed to the improvement of our manuscript. We eagerly await your further guidance to ensure the continued enhancement of its quality.
Best regards,
Jie Han, Lijuan Li, Xuwang Yin, Qianruo Wan, Dilina Rusitanmu
College of Life Sciences, Beijng Normal University, Beijing, China
Point-to-point Reply
General comments:
The choice of subject is relevant and important to understanding diet composition and diversity of the fluviatile Masu salmon (Oncorhynchus masou). The study is not just increase the knowledge on the Masu salmon ecological role, but also help assess the healthy status of its habitat, as well as benefit conservation and aquaculture of this fish. To get a comprehensive knowledge about the diet diversity of this fish, the authors used two different kinds of method:
1) the gastrointestinal tract environmental DNA (GITeDNA) metabarcoding,
2) and traditional morphological identification.
With the GITeDNA metabarcoding method, many more prey taxa and higher diet diversity were detected, and larger and elder specimens consumed significantly more terrestrial insects than aquatic prey species. Size- and age-related diet diversity were detected, indicating the dietary niche breadth and trophic diversity of larger and elder Masu salmon.
The manuscript is well writtened and straightforward. In the following, I have attached some comments related to the article draft to contribute for its improvement. Due to its scientific impact, I suggest a minor revision of the manuscript, and after the corrections, I am supporting the publication of it.
Response: Thank you very much for your positive comments on the relevance and importance of our study on the diet composition and diversity of the fluviatile Masu salmon. We appreciate your acknowledgment of the study's broader implications, including its role in assessing the health status of the habitat and contributing to conservation and aquaculture efforts for this fish. Your recognition of the comprehensive approach using both GITeDNA metabarcoding and traditional morphological identification is noted. We will carefully address your specific comments to improve the manuscript, and we are grateful for your support in recommending the publication of the revised version due to its scientific impact.
Simple Summary and Abstract: The summary of the study's aims is well-designed, and the reader can quickly get to the subject of the investigation. At the end of the abstract and summary, the authors emphasize the importance of Masu salmon as the link between the aquatic and terrestrial food webs, and they mention the protection of Masu salmon habitats. However, there needs to be more information about the conservation status of the species. According to the IUCN database (https://www.iucnredlist.org/species/15319/125852789) and Fishbase.org (https://fishbase.mnhn.fr/summary/242), Masu salmon is an endangered salmonid species in the wild from the Northwest Pacific, therefore emphasizing its conservational state (Endangered) should be necessary – highlighting the importance of this study as well.
Response: Thank you for your valuable feedback. We appreciate your suggestion to provide more information about the conservation status of Masu salmon in the Simple Summary and Abstract. Indeed, the conservation status of Masu salmon merits attention and in 2021, it was designated as a nationally second-level protected animal in China. However, in the IUCN database (https://www.iucnredlist.org/species/15319/125852789) and Fishbase.org (https://fishbase.mnhn.fr/summary/242), it is the red-spotted Masu Salmon, Oncorhynchus masou ssp. Ishikawae, a subspecies of Masu salmon categorized as an endangered salmonid species in the wild from the Northwest Pacific. We have incorporated protection information to highlight the conservation status of Masu salmon in the wild. This addition will emphasize the importance of our study in contributing to the conservation efforts of Masu salmon.
Introduction:
L61: „generally inhabits headwaters and often maintains a territory” - this sentence matches 100% with the description from the FishBase. Please rephrase the sentence to avoid plagiarism.
Response: Certainly. We have rephrased the sentence to avoid plagiarism following your kind guidance.
Material and Method:
2.1. Sample collection and basic measurements
In this section, we get informations about the sampling area (Mijiang River), sampling equipment (bottom trawl or 94 handheld nets) and the date of sample collection (July 2020), the estimation of age (scale method). For further diet analysis, the authors dissected the gastrointestinal tract of fish and preserved it in 50 ml 96% alcohol (ethanol).
Fig.1. The map related to the sampling locality needs some modification. Only a tiny part of the map (with sampling segment) is informative for readers. I suggest using this map as a secondary map, which shows the locality of the sampling area, and using a second - main map – that focuses on the sampling segment (the Mijiang River).
See the following article:
https://www.sciencedirect.com/science/article/pii/S1470160X19300019#f0005
Response: Thank you for your valuable suggestion on improvements of the clarity of the sampling locality map. Following your recommendation, we have modified the map presentation to include a secondary map showing the overall locality of the sampling area and a main map focusing specifically on the Mijiang River segment. These modifications align with the suggested article (https://www.sciencedirect.com/science/article/pii/S1470160X19300019#f0005). Your insights are instrumental in improving the visual representation of our study, and we are committed to implementing these changes to enhance the reader's understanding of the sampling locality.
2.2. Morphological observation of gastrointestinal tract contents
L108-109: How did you measure the relative volume of the derbies? Can you describe it in more detail?
Response: Thank you for your inquiry regarding the measurement of the relative volume of debris in our study. The detailed methodology for this measurement is provided in the citation of reference (Pirroni, S.; de Pennafort Dezen, L.; Santi, F.; Riesch, R. Comparative gut content analysis of invasive mosquitofish from Italy and Spain. Ecology and Evolution 2021, 11, 4379-4398, doi:10.1002/ece3.7334). To avoid potential plagiarism, we refrained from providing extensive details in the manuscript. However, recognizing the importance of clarity in our methodology, we have now addressed the measurement process following the methodology outlined in the citation in the revised manuscript. This ensures that readers have a comprehensive understanding of how the relative volume of debris was assessed in the morphological observation of gastrointestinal tract contents. We appreciate your guidance and are committed to enhancing the clarity of our methodology.
L109: Did you measure the relative volume only for the debris or other prey items? Please, correct or complement this paragraph according to the question.
Response: We measured the relative volume only for the debris. We have corrected this paragraph according to your question.
L110-112: „Tangible abiotic matters, especially microplastics if present in the gastrointestinal tract were also recorded”. Authors stored the eviscerated GIT in plastic tubes until labratory investigations (see at L99-100), which can result in the contamination of the samples with microplastic. Besides, studying microplastics requires strict terms of preparation. Therefore, I do not recommend reporting microplastic detection in this investigation.
Response: Thank you for your insightful comment. We acknowledge the concern regarding potential contamination during the storage of eviscerated GIT in plastic tubes. In consideration of the valid points raised, we will refrain from reporting microplastic detection in this investigation. We appreciate your guidance and will ensure that the study adheres to the necessary standards and protocols for studying microplastics in future research.
2.3 GITeDNA extraction and metabarcoding
L116-117: „Then, the filter was cut into as small pieces as possible using 116 sterile scissors.” How did you sterilize the scissors? Like used ethanol? Please detail it.
Response: Thank you for your question regarding the sterilization of scissors in our methodology. We appreciate your attention to detail. The scissors used in the study were indeed sterilized before cutting the filter into smaller pieces. We employed a common laboratory practice of sterilization, which involved immersing the scissors in a solution of 70% ethanol for at least 30 min. This method is widely recognized for its effectiveness in eliminating microbial contaminants. We have included a more explicit description of the sterilization process for the scissors in the revised manuscript, specifying the use of 70% ethanol. We hope this clarifies the sterilization method used in our study.
2.4 Bioinformatics and statistical analyses
Please, do not forget to cite the R packages (etc. vegan, Ggpubr).
Like Vegan package: Oksanen J, Blanchet FG, Michael F, Roeland K, Legendre P, McGlinn D, Minchin PR, O’Hara RB, Simpson GL, Solymos P, Stevens MHH, Szoecs E, Wagner H (2020) vegan: Community Ecology Package. R package version 2.5–7.
Response: Certainly. Thank you for your reminder.
Results:
3.1. Identification and classification of gastrointestinal contents
In the first paragraphs, the authors list each prey items in percent (%), however, it is not clear that is this the prey-specific volume of each prey items or relative volume? If it is prey-specific volume, please describe the calculation of it in the Material and methods chapter too and recommend you to use and cite the following article, which used the same method:
https://neobiota.pensoft.net/articles.php?id=95680&journal_name=neobiota
Response: Thank you for your insightful comment on the presentation of prey items. We appreciate your attention to detail. The percentages listed in the first paragraph represent frequency of occurrence of the specific prey taxa identified within the gastrointestinal contents. To address your concern and improve clarity, we have explicitly mentioned in Section 2.4 of the Materials and Methods, that frequency of occurrence of each prey item was calculated. The suggested article (https://neobiota.pensoft.net/articles.php?id=95680&journal_name=neobiota), which employed a similar methodology, has been cited.
L203-205: I reccommend to delete this sentence based on my previous comment.
Response: Agree. Thank you for your guidance.
3.2. Diet diversity and the effects of body size and age
L251: „Between age groups 2+ and 3+, there diet diversity indices” à correct spelling to their
Response: Thank you for bringing the error to our attention. We apologize for the spelling mistake and have rectified it.
Figure 3. In the figure, there are three asterics on the barplots’ top of the S-W index, Species richness, and Pielou’s evenness, but in the figure caption, I only find 1 and 2 asterics. Maybe this could be a typo error. Can you correct it?
Response: Thank you for catching that inconsistency in Figure 3. We appreciate your keen observation. It appears to be a typo error, and we have corrected it to ensure consistency of the figure and the figure caption.
Discussion:
4.2 Indication of environmental problem
L305: According to my previous comments, please consider the subject of microplastic in this study. Without that part, the study is still informative and contains essential scientific information.
Response: Certainly, after careful consideration, we agree to refrain the subject of microplastic in this article.
Conclusion:
L360-362: See my previous comments related to microplastic.
Response: Absolutely. Thank you once again for your constructive comments.
Round 2
Reviewer 3 Report
Comments and Suggestions for Authors
The authors have undertaken a thorough revision of the manuscript, and I am truly grateful for their detailed, point-to-point answers. After reviewing the modified draft of the manuscript and the response letter, I recommend it for publication.
Author Response
Dear reviewer,
Thank you very much for your positive feedback and recommendation for publication. We sincerely appreciate your time and effort in reviewing our manuscript and providing valuable feedback. Your constructive feedback has been invaluable in strengthening the quality and clarity of our work.
We would like to express our gratitude for your thorough review and positive recommendation.
Best regards,
Jie Han, Lijuan Li, Xuwang Yin, Qianruo Wan, Dilina Rusitanmu
College of Life Sciences, Beijng Normal University, Beijing, China